# Urinary L-FABP as an Early Biomarker for Pediatric Acute Kidney Injury Following Cardiac Surgery with Cardiopulmonary Bypass: A Systematic Review and Meta-Analysis

**DOI:** 10.3390/ijms25094912

**Published:** 2024-04-30

**Authors:** Bruno Wilnes, Beatriz Castello-Branco, Bárbara Castello Branco, André Sanglard, Pedro Alves Soares Vaz de Castro, Ana Cristina Simões-e-Silva

**Affiliations:** Interdisciplinary Laboratory of Medical Investigation, Unit of Pediatric Nephrology, Department of Pediatrics, Faculty of Medicine, Federal University of Minas Gerais, Belo Horizonte 30130-100, MG, Brazil; brunowilnes@ufmg.br (B.W.); beatrizcastellob@ufmg.br (B.C.-B.); barbaracastello@ufmg.br (B.C.B.); ads0311@ufmg.br (A.S.); pedroasvc@gmail.com (P.A.S.V.d.C.)

**Keywords:** acute kidney injury, cardiac surgery, pediatrics, cardiopulmonary bypass, biomarkers, liver-type fatty acid-binding protein

## Abstract

Acute kidney injury (AKI) following surgery with cardiopulmonary bypass (CPB-AKI) is common in pediatrics. Urinary liver-type fatty acid binding protein (uL-FABP) increases in some kidney diseases and may indicate CPB-AKI earlier than current methods. The aim of this systematic review with meta-analysis was to evaluate the potential role of uL-FABP in the early diagnosis and prediction of CPB-AKI. Databases Pubmed/MEDLINE, Scopus, and Web of Science were searched on 12 November 2023, using the MeSH terms “Children”, “CPB”, “L-FABP”, and “Acute Kidney Injury”. Included papers were revised. AUC values from similar studies were pooled by meta-analysis, performed using random- and fixed-effect models, with *p* < 0.05. Of 508 studies assessed, nine were included, comprising 1658 children, of whom 561 (33.8%) developed CPB-AKI. Significantly higher uL-FABP levels in AKI versus non-AKI patients first manifested at baseline to 6 h post-CPB. At 6 h, uL-FABP correlated with CPB duration (r = 0.498, *p* = 0.036), postoperative serum creatinine (r = 0.567, *p* < 0.010), and length of hospital stay (r = 0.722, *p* < 0.0001). Importantly, uL-FABP at baseline (AUC = 0.77, 95% CI: 0.64–0.89, n = 365), 2 h (AUC = 0.71, 95% CI: 0.52–0.90, n = 509), and 6 h (AUC = 0.76, 95% CI: 0.72–0.80, n = 509) diagnosed CPB-AKI earlier. Hence, higher uL-FABP levels associate with worse clinical parameters and may diagnose and predict CPB-AKI earlier.

## 1. Introduction

Acute kidney injury (AKI) is a common and life-threatening condition among critically ill children, with some reports indicating mortality rates between 11% and 64% for pediatric patients requiring kidney replacement therapy [1,2]. Despite a variety of conditions that cause AKI, children, especially infants, submitted to cardiac surgery under cardiopulmonary bypass (CPB) are at increased risk of developing kidney failure. Accordingly, in a cohort including only infants, 56.1% of the patients submitted to CPB developed AKI within five days of surgery [3].

Although not yet completely understood, AKI after CPB (AKI-CPB) presents a very unique pathophysiology. The current understanding about the pathogenesis of the condition states that AKI-CPB is anchored in the concomitant occurrence of, at least, four main processes: reduced kidney perfusion pressure, hemolysis, activation of proinflammatory pathways, and formation of microemboli [4]. Firstly, CPB exposes the patients’ kidneys to a decrease of up to 30% in its perfusion pressure, a fact that, alongside CPB-induced hemolysis, favors ischemic damage to the kidney parenchyma [5]. Simultaneously, CPB inherently promotes the development of microemboli, formations that are small enough to evade the bypass filters and damage kidney capillaries4. Finally, these pathological phenomena, in association with the systemic inflammatory response experienced by the CPB patient, trigger ischemia-reperfusion injury of the kidneys, potentially leading to AKI-CPB [4].

The definition of AKI has, since 2012, been unified by the Kidney Disease Improving Global Outcomes (KDIGO) Acute Kidney Injury Work Group through the KDIGO Criteria as a predefined increase in serum creatinine (SCr) or decrease in urine output [6]. As stated previously, oliguria is an important parameter for the identification of AKI. In addition, since oliguria is the clinical repercussion of an already symptomatic AKI, this sign is inherently late. Despite being a very cost-effective and available biomarker, SCr is also not able to early detect AKI. The findings suggest that SCr might only indicate AKI-CPB in pediatric patients 48 h after the beginning of the injury [7,8].

For these reasons, the assessment of new biomarkers that early detect AKI-CPB emerges as a critical strategy for improving the management of this set of patients. More specifically, the assessment of the molecule in urinary samples has some practical advantages in comparison to serum ones, especially when considering children submitted to cardiac surgery under CPB. Most of these patients, due to the expectancy of long postoperative ICU admission, will have a urinary catheter, which facilitates urine collection, even for serial assessments, and makes it less invasive than the obtention of serum samples.

In this context, the Liver-type Fatty Acid-binding Protein (L-FABP), a 14 kDa protein from the superfamily of lipid-binding proteins, emerges as a promising biomarker for the prediction, diagnosis, and prognosis assessment of kidney function in patients with AKI post-CPB. In the kidneys, the L-FABP is mostly expressed in the proximal tubules, where it is responsible for the transportation of fatty acids into the tubular epithelial cells’ mitochondria to be converted into energy [9,10]. Especially in the kidneys, the L-FABP gene has been found to be upregulated during hypoxic stress and ischemia-reperfusion injury, two major mechanisms involved in the pathogenesis of AKI-CPB [9,10]. During a hypoxic event, the damage of the proximal tubules is largely caused by the accumulation of reactive oxygen species (ROS) in the kidneys. This process is inhibited by L-FABPs’ capacity to bind with lipid peroxidation products and induce the excretion of these molecules in the urine, being detected as urinary L-FABP (uL-FABP) [10]. Therefore, evidence indicates that uL-FABP undergoes a compensatory increase very early in hypoxia-mediated AKI.

In this regard, a prospective study with 1273 cardiac intensive care unit (ICU) patients, with a mean age of 68 years, identified urinary uL-FABP as an independent predictor of AKI, and the use of this biomarker was able to improve early diagnosis of AKI [11]. Therefore, the assessment of L-FABP levels, commonly conducted through the Enzyme-Linked Immunosorbent Assay (ELISA) technique, emerges as a potential strategy for the prediction and early diagnosis of AKI post-CPB.

Hence, this systematic review and meta-analysis aimed to evaluate the potential value of uL-FABP in the early diagnosis and prediction of AKI among infants submitted to cardiac surgery with cardiopulmonary bypass.

## 2. Methods

### 2.1. Protocol Design and Registration

The research protocol was registered in the International Prospective Register of Systematic Reviews (PROSPERO), identified as CRD42022318748, under the title “Biomarkers’ effectiveness to define diagnosis, prognosis and severity of pediatric acute kidney injury in Emergency Units: a systematic review”. The registration can be found in (https://www.crd.york.ac.uk/prospero/display_record.php?ID=CRD42022318748, accessed on 11 November 2023). Throughout the entire development process of this review, adherence to the Preferred Reporting Items for Systematic Reviews and Meta-Analyses [12] (PRISMA) recommendations was rigorously maintained.

### 2.2. Eligibility Criteria

Observational studies employing a case-control, cohort, or cross-sectional design, as well as diagnostic test accuracy studies, were included in this systematic review. The focus was on assessing the capability of uL-FABP for early detection of post-surgical AKI, and prediction of severe AKI development, in pediatric patients (under 19 years of age) undergoing cardiac surgery with CPB. Articles published in English, Spanish, French, and Portuguese were considered for appropriate screening. Exclusion criteria were: studies including only an adult population (older than 19 years), studies that did not analyze the development of AKI, studies with patients solely undergoing surgery without CPB, and studies that did not align with the research question.

### 2.3. Information Sources and Search Strategies

The search strategy was designed using the entry terms “Acute Kidney Injury”, “Cardiac Surgery”, “Cardiopulmonary Bypass”, “Liver Fatty Acid-Binding Protein”, “Urine”, “Children”, and similar keywords extracted from the Medical Subject Headings (MeSH) 12 November 2023. A systematic search of the literature was performed in the databases PubMed/MEDLINE, Web of Science, Scopus, and Cochrane. The complete search strategy can be found in Appendix A.

To ensure a thorough and comprehensive systematic search, the gray literature was explored through both reference analysis of selected articles and access to Google Scholar. The search was updated as of the 15 March 2024 to incorporate any relevant updates or newly published materials.

### 2.4. Study Selection and Data Extraction

Following the systematic search and removal of duplicate records, six independent researchers proceeded to screen articles based on their titles and abstracts, obeying the predefined inclusion and exclusion criteria. Subsequently, the remaining articles underwent thorough reading in their entirety as the final step of the screening process. All papers that met the inclusion criteria were comprehensively reviewed.

Data extraction comprised the following variables: authorship, year, number of participants and grouping characteristics, study design, gender, age, characterization of data collection, inclusion and exclusion criteria, AKI definition criteria, and clinical outcomes (progression to dialysis, and death). Clinical and laboratory characteristics were also extracted and included: AKI stage stratification, prior CPB and CPB duration, pre-operatory risk adjustment in congenital heart surgery (RACHS) score, and uL-FABP values according to group and time after CPB. Finally, we also performed the extraction of synthesis metrics: area under the ROC curve (AUC) for the accuracy of AKI detection by uL-FABP, and correlation coefficients between uL-FABP levels and relevant clinical parameters.

### 2.5. Methodological Quality Evaluation of the Included Studies

Two researchers independently assessed the methodological quality of the included studies. We employed the JBI Critical Appraisal Checklist to evaluate methodological quality. Studies were assessed by the JBI Critical Appraisal Checklist for Diagnostic Test Accuracy Studies [13] or by the JBI Critical Appraisal Checklist for Case-Control Studies [14], according to study design. The detailed quality assessment process can be found in Appendix A.

### 2.6. Statistical Analysis

The study data were extracted and organized using Microsoft Excel. In instances where statistical synthesis posed significant limitations, we opted for a narrative or graphical synthesis approach. Studies that reported the AUC for the accuracy of urinary L-FABP levels in detecting AKI were aggregated through a generic inverse variance multivariate meta-analysis. When original manuscripts lacked the 95% confidence intervals (95% CI) and standard error (SE) of AUC values, we computed the estimated 95% CI and SE utilizing the total number of included patients and the AUC values from each study individually. The SE and the 95% CI were estimated as described by Fisher et al. (1997) [15]. Both random and fixed/common models were evaluated and presented. Cochran’s Q-test and Higgins’ I^2^ test were computed for each model. All data were presented as effect estimates with 95% CI. Analyses were conducted using the “meta” and “metafor” R statistical packages in R Studio, version 4.3.3, R Foundation for Statistical Computing, Vienna, Austria.

## 3. Results

### 3.1. Studies Design

Nine observational studies were included (Figure 1), published from 2008 to 2018 [16,17,18,19,20,21,22,23,24]. In total, 1658 patients were analyzed. Seven studies were prospective [16,17,19,20,21,23,24], and two were case-control studies [18,22]. Sample sizes varied among the studies and ranged from 27 to 408 participants. Eight studies divided patients according to their gender [16,18,19,20,21,22,23,24]: out of 1250 patients, 638 (51.0%) were boys and 612 (49.0%) were girls. Characteristics of location, interventions, and sample sizes of the included studies are described in Table 1.

Inclusion criteria were similar between studies and comprised pediatric patients submitted to cardiac surgeries with the use of CPB. Exclusion criteria were disclosed for 8 articles and varied according to the paper, but severe and chronic kidney disease (CKD) was frequently reported: 7 of 8 studies excluded patients with CKD or kidney insufficiency [16,18,19,21,22,23,24]; 2 studies excluded patients with congenital abnormalities of the kidney and urinary tract (CAKUT) [16,18]; 3 studies excluded patients who were taking nephrotoxic drugs [16,18,19].

The definition of AKI also differed among studies. Seven articles utilized the KDIGO parameters of AKI, defined as a 50% increase or an absolute increase of 0.3 mg/dL in SCr levels [17,18,19,21,22,23,24]. One article defined AKI as a 25% decrease of the glomerular filtration rate (GFR), estimated by Schwartz’s formula [16]. One final paper defined AKI as the doubling of SCr or the need for acute dialysis [20]. Finally, one paper divided AKI patients between two groups, according to disease progression [17]. AKI with progression was defined as a progression in the AKI stage, or as a persisting stage 3 AKI, for at least 2 consecutive days. Baseline results were only evaluated when they were defined as the time immediately after surgery or at post-CPB ICU admission. In all the studies, L-FABP measurements were consistently conducted using the Enzyme-Linked Immunosorbent Assay (ELISA) technique.

The primary outcome was the development of post-CPB AKI in 8 of the 9 (88.9%) included studies [15,17,18,19,20,21,22,23], with one study evaluating AKI progression as the primary outcome [16]. None of the included studies clearly defines the secondary outcomes analyzed. However, since clinically relevant outcomes other than AKI development (length of hospital stay, progression to dialysis, and death) were reported, these results were also qualitatively analyzed in the Section 3.

### 3.2. Population Data

Out of the 1658 patients, 561 (33.8%) developed AKI after CPB. The prevalence of AKI varied from 16.0% to 52.5% in the studies. In only one study, the percentage of patients who developed AKI was over 50% [19]. All studies divided AKI and non-AKI patients according to their sex. A total of 273 (44.6%) girls evolved with AKI versus 288 (45.1%) boys who developed the condition. In all articles, there was no significant difference in AKI occurrence between genders. The age of patients who developed AKI varied from 3.84 (0.36–28.8) months to 60.24 ±10.92 months. In non-AKI patients, the age varied from 7.2 (4.8–21.6) months to 57.6 ±56.4 months. Five studies [20,21,22,23,24] revealed significant age difference between patients with and without AKI. Parikh’s et al. [20], Zappitelli’s et al. [23], and Yoneyama et al. [24] studies showed a notably lower age for the AKI group when compared to the non-AKI group; conversely, Krawczeski et al. [21] and Dong et al. [22] reported higher ages for the AKI group when compared to the non-AKI group, the ages being, respectively, 7.2 (4.8–21.6) months versus 17.4 (2.4–32.4) months. Details of baseline characteristics of the population are described in Table 2.

### 3.3. Cardiopulmonary Bypass Duration

All nine studies compared the bypass duration between the AKI and non-AKI groups and showed statistically significant differences. Patients who developed AKI had a mean time of bypass that ranged from 113 (84–172) to 240 (183–297) minutes. In comparison, children without AKI had a time of bypass from 82 ± 9.7 to 126 ± 65 min.

### 3.4. RACHS Score

The RACHS score was assessed in 8 studies [16,17,18,20,21,22,23,24], of which 7 compared results between the AKI and non-AKI groups. In these 7 studies, 364 out of 1210 (30.0%) patients developed AKI. The classification of RACHS scores varied between the AKI and non-AKI groups and was available for 1208 patients. A RACHS score of ≤3, indicating a lower complexity of cardiac surgery, was more commonly observed in the non-AKI group (95.8%) as compared to the AKI group (89.2%) in these seven studies.

### 3.5. Clinical Outcomes

The need for dialysis was assessed by 3 studies [16,17,18] that comprised 547 children. Of those, 10 (1.8%) patients required dialysis. Only 2 studies compared the need for this type of kidney replacement therapy between AKI and non-AKI groups [16,18]. In these papers, the difference between the AKI and non-AKI groups was statistically significant, and all 7 patients who needed dialysis belonged to the AKI group, corresponding to 24.1% of all AKI patients. In the third study [17], the need for dialysis was only compared between two AKI groups: AKI with progression and AKI without progression. All three patients requiring dialysis were in the group with disease progression, reaching a statistically significant difference between the groups. Additionally, only one study combined dialysis and death as a single composed outcome [20]. In this study, 9 out of 303 (3.0%) children progressed to the composed outcome, with no statistically significant difference between groups (*p* = 0.19).

The progression to death, alone, was analyzed in 4 studies [16,17,18,23], with a total of 834 children; of those, 14 (1.7%) died. Only 2 studies compared the progression to death between the AKI and non-AKI groups [16,18]. In one study, all four deaths occurred among patients with AKI, representing 16.7% of the AKI group, reaching a statistically significant difference (*p* < 0.0001). However, in the second study, two deaths occurred: one among AKI patients and one among non-AKI patients, indicating no significant difference between the groups. In the third study [17], the comparison focused solely on the need for dialysis between two AKI groups: AKI with progression and AKI without progression. However, there was no statistically significant difference in the occurrence of death between the two groups.

### 3.6. Urinary L-FABP

Six studies described levels of uL-FABP in patients with and without AKI at various time points after CPB surgery [17,19,20,21,22,24]. Five of these studies compared uL-FABP levels between patients who developed AKI and those who did not [19,20,21,22,24]. The remaining study focused on comparing uL-FABP levels between AKI patients who showed disease progression and those who did not [17].

At baseline, one study [24] revealed significantly elevated levels of uL-FABP among patients with AKI compared to those without AKI (*p* < 0.05). At 2 h post-CPB, two independent studies reported higher uL-FABP levels in the AKI group [16,18]. Nonetheless, this difference was not observed in two other studies [21,22]. At 4 h post-CPB, two separate studies [19,24] indicated significantly higher levels of uL-FABP in the AKI group compared to the non-AKI group (*p* < 0.05; *p* < 0.01). Similar results were observed at 6 h post-CPB when four separate studies [16,18,21,22] reported elevated levels of uL-FABP in the AKI group compared to the non-AKI group (*p* < 0.05; *p* < 0.0001; *p* < 0.05; *p* < 0.05). At the 12-h timepoint, four distinct studies [19,21,22,24] consistently reported substantially elevated levels of uL-FABP in the AKI group compared to the non-AKI group. Finally, at 24 h post-CPB, four studies identified higher levels of uL-FABP in the AKI group [16,21,22,24]. However, one study reported no difference between the two groups at this time point [18].

Two studies identified higher uL-FABP levels in the AKI group in the time interval of 0 to 6 h post-CPB (*p* < 0.05) [20,23] and one study also identified this difference in the time interval of 6 to 12 h post-CPB [20]. One study [17] compared uL-FABP levels between AKI patients who showed disease progression and those who did not and pointed out that elevated levels of uL-FABP were significantly higher in AKI patients with disease progression (*p* < 0.05). The peak uL-FABP values, i.e., the highest value in 24 h, varied among the studies, occurring at different timepoints ranging from 4 to 12 h after the procedure. The values at the peak also exhibited significant variation, with reported values ranging from 264.76 (175.90–398.50) ng/mL to 791 ± 349 ng/mL.

### 3.7. Correlations of uL-FABP with Relevant Clinical Parameters

The reviewed studies also established several correlations between uL-FABP levels and various clinical parameters of the included patients. The length of hospital stay showed significant correlations with uL-FABP levls at 4 h [19] (r = 0.57825; *p* = 0.0017), 6 h [16,18,21] (r = 0.722; *p* < 0.0001; er = 0.248, *p* < 0.05 and r = 0.37; *p* < 0.05), 12 h [19] (r = 0.53568; *p* = 0.0004), and 24 h [16] (r = 0.424; *p* < 0.05) post-CPB.

The percentage change in postoperative serum creatinine correlated with uL-FABP levels at 2 h [16,18] (r = 0.486; *p* < 0.01; r = 0.680, *p* = 0.001), 4 h [19] (r = 0.465; *p* = 0.0025), 6 h [16,21] (r = 0.567; *p* < 0.01; r = 0.33, *p* < 0.05), 12 h [19] (r = 0.479; *p* = 0.0017), and 24 h [16] (r = 0.466; *p*< 0.05) post-CPB. Furthermore, urinary L-FABP also demonstrated correlations with CPB time at 2 h [16,18] (r = 0.509; *p* < 0.05; r = 0.408, *p* = 0.045) and 6 h [18,21] (r = 0.498, *p* = 0.036; r = 0.33, *p* < 0.05), aortic cross-clamping time at 2 h [16,18] (r = 0.650; *p* < 0.01; r = 0.536, *p* = 0.032), and the Risk Adjustment for Congenital Heart Surgery (RACHS) score at 2 h [16] (r = 0.785; *p* < 0.01) and 6 h [21] (r = 0.21, *p* < 0.05) post-surgery.

### 3.8. Accuracy of uL-FABP in Early Diagnosis of Post-CPB AKI

All nine studies [16,17,18,19,20,21,22,23,24] examined the potential of uL-FABP levels for early diagnosis of AKI and prediction of severe AKI development at various post-CPB timepoints. Ivanišević [18] and colleagues pointed out that, when combining the uL-FABP levels with a clinical model comprised by age, sex, weight, CPB time, and aorta clamp time, the diagnostic ability of the model was improved to an AUC of 0.94 at 2 h, 0.99 at 6 h, and 0.97 at 24 h post-CPB [18]. The early diagnostic ability of uL-FABP combined with the clinical model was higher than that of the clinical model alone. Similarly, Krawczeski [21] and colleagues pointed out that, when combined with a clinical model including age and CPB time, uL-FABP levels diagnosed AKI development with an AUC of 0.73 at 2 h, 0.77 at 6 h, 0.81 at 12 h, and 0.79 at 24 h. However, combining these parameters did not improve early diagnostic ability beyond that of the clinical model alone.

Moreover, Portilla and colleagues [19] identified that, at 4 h post-CPB, uL-FABP levels could diagnose, alone, the development of AKI with an AUC of 0.81 (*p* = 0.0265). Two studies [20,23] showed that uL-FABP levels were able to diagnose AKI with AUCs of 0.70 ± 0.04 and 0.65 (0.58–0.71) in the time interval of 0 to 6 h post-CPB and with an AUC of 0.71 ± 0.04 in the time interval of 6 to 12 h post-CPB [20]. Furthermore, uL-FABP levels were also found to be predictive of AKI progression, demonstrating an optimism-corrected AUC of 0.71 [17] (0.6, 0.8). The detailed AUCs of uL-FABP to diagnose AKI at different post-CPB timepoints are described in Table 3.

In the meta-analysis of pooled AUC results [16,18,19,21,22,24] uL-FABP was able to diagnose AKI at five different time points: 2 h, 4 h, 6 h, 12 h, and 24 h after CPB (Figure 2 and Figure 3), with 24 h presenting the highest accuracy (AUC = 0.80, 95% CI: 0.74–0.86), as shown in Figure 3 [16,21,22,24]. Although less accurate, uL-FABP still presented satisfactory results at 2 h [16,18,21,22] (AUC = 0.71, 95% CI: 0.52–0.90; n = 509) and 4 h [19,24] (AUC = 0.79, 95% CI: 0.72–0.86; n = 143), as shown in Figure 2. Significant results were also found at 6 h [16,18,21,22] (AUC = 0.76, 95% CI: 0.72–0.80; n = 509) and 12 h [21,22,24] (AUC = 0.76, 95% 0.64–0.88; n = 473), as also displayed in Figure 3.

### 3.9. Quality Assessment

As for the quality assessment, diagnostic test studies (n = 5) were evaluated by the JBI Critical appraisal checklist for diagnostic test accuracy studies [13]. Five studies [16,19,20,23,24] were considered of good quality and two [17,21] were considered of fair quality. The main defects were inappropriate exclusion (identified in two papers [17,21]), unclear information regarding whether the analysis of the reference standard test was blinded to the index test results (n = 6) [17,19,20,21,23,24], and unclear information of whether the index test was interpreted blinded to the reference standard results (n = 1) [21]. Case-control studies (n = 2) [18,22] were evaluated using the JBI Critical appraisal checklist for case-control studies [14]. The studies were assessed as of good quality, but confounding factors and strategies to deal with those factors were not stated in either of them. The complete quality assessment can be found in Appendix A.

## 4. Discussion

AKI is a prevalent issue among pediatric patients undergoing cardiac surgery involving cardiopulmonary bypass (CPB), with an estimated incidence rate of 40% to 50% of all pediatric CPB procedures [7]. In this sense, the early identification of AKI is crucial to improve patient outcomes within this specific group. However, the existing diagnostic criterion for AKI, which relies on SCr measurement and urinary output, is rather late. Consequently, the evaluation of novel biomarkers for AKI post-CPB has emerged as a promising strategy to enable early identification and subsequent treatment for these patients. Several biomarkers have been investigated for the early detection of AKI after surgeries with CPB. Our review focuses on uL-FABP obtained until 24 h postoperatively.

A recent meta-analysis of diagnostic test accuracy has addressed the potential value of a group of biomarkers, including L-FABP, in serum and urinary samples of children submitted to cardiac surgery [25]. However, our approach differs in some respects from this previous study. First, our population is comprised exclusively of infants submitted to cardiac surgery with CPB; second, our approach focuses on levels of L-FABP in urinary samples only, not including serum measurements; and third, we focus on the early analysis of the biomarker, including data no later than 24 h post-CPB.

Our data reinforce that, in the early hours after the kidney insult, uL-FABP levels are significantly increased in patients who will develop AKI. In our review, in patients that developed post-CPB AKI when compared to the non-AKI group, uL-FABP levels were already significantly increased at baseline [24] and were consistently higher in the AKI group, at 4-, 6-, 12-, and 24-h post-surgery [16,18,19,21,22,24]. Additionally, uL-FABP was found to reach peak concentration in the urine shortly after CPB, between 2- and 12-h post-surgery [16,18,19,20,21,22], a finding that is in line with the capacity of uL-FABP to early indicate an increased risk of developing post-CPB AKI. Interestingly, uL-FABP was also significantly increased in patients who developed AKI stages 2 or 3 when compared to patients who exhibited AKI stage 1 or no AKI [22]. Despite only one study addressing this type of direct comparison, this finding indicates the potential for uL-FABP to act both as an early biomarker for diagnosis and severity of post-CPB AKI.

Levels of uL-FABP significantly correlated with various clinical parameters of severity, including length of hospital stay, percentage of change in postoperative serum creatinine, and pre-operative RACHS score. Correlations were also observed between uL-FABP levels and procedural parameters, such as CPB time and aortic cross-clamping time. Despite most correlations being moderate, our data support the intrinsic relationship between uL-FABP and the magnitude of the acute insult to the kidneys, which ultimately results in AKI.

Our systematic review and meta-analysis suggest that early uL-FABP exhibits satisfactory accuracy in both the early diagnosis of post-CPB AKI and the prediction of progression in the CPB-AKI stage in children. In this matter, albeit assessed in few articles, uL-FABP obtained 2 h after surgery showed to be satisfactorily accurate to diagnose post-CPB AKI [16,18]. Urinary L-FABP was also valuable for the prediction of AKI progression post-CPB [17]. As a general tendency, the predictive and diagnostic capacity of uL-FABP increased in later post-CPB timepoints and reached its peak at 24 h post-surgery, which is significantly earlier than the traditional biomarkers. The meta-analysis data reinforce this trend but, notably, uL-FABP alone presented significant and favorable accuracy in post-CPB timepoints as early as two hours after surgery. In addition, although accuracy increased in later timepoints, the meta-analysis shows that earlier post-CPB uL-FABP measurements had high accuracy and were not significantly different from the diagnostic capacity observed at the 24-h threshold. Importantly, although only assessed by one study, uL-FABP associated with a robust clinical model presented excellent accuracy in diagnosing AKI-CPB at very early post-surgery timepoints. Despite models comprising both uL-FABP and clinical parameters not resulting in a similarly high AUC in other papers, the association between this urinary biomarker and a robust clinical model may be a cost-effective way to reach consistent early detection of post-CPB AKI.

Although our systematic review and meta-analysis highlight important considerations for the use of uL-FABP as a biomarker of post-CPB AKI in pediatric patients, our analysis harbors some limitations. First, as mentioned above, the limited number of studies using the same definitions for AKI increases the heterogeneity of the assessed clinical outcomes and hampers a more in-depth meta-analysis. Furthermore, the absence of some critical information in the manuscripts for the performance of the meta-analysis unfortunately reduced the overall number of included studies. This is especially true for the accuracy of uL-FABP at 4 h post-CPB, which was only evaluated in two studies. Additionally, despite the AUC consisting in an important and reliable metric for the evaluation of a biomarker’s diagnostic and predictive capacity, an in-depth assessment of a diagnostic test also relies on the analysis of statistical properties associated with a fixed cut-off value, such as accuracy and likelihood ratios. Since the included studies did not establish cut-off values for the early diagnosis of post-CPB AKI, these metrics were not evaluated. Moreover, despite the overall good quality of the included studies, possible sources of bias were identified. In that sense, larger and well-designed studies are still needed to better evaluate the use of uL-FABP as an early diagnostic and prognostic biomarker in post-CPB AKI.

## 5. Conclusions

In conclusion, uL-FABP emerges as a promising early biomarker for AKI following CPB in pediatric patients. Despite study limitations, its correlation with AKI development and severity suggests its potential for enhancing timely intervention and improving patient outcomes. Moreover, the results of the meta-analysis support that uL-FABP levels are capable of diagnosing AKI after CPB with good accuracy, especially after 24 h of surgery. Nevertheless, further research is needed to standardize criteria and validate the clinical utility of uL-FABP.

## Figures and Tables

**Figure 1 ijms-25-04912-f001:**
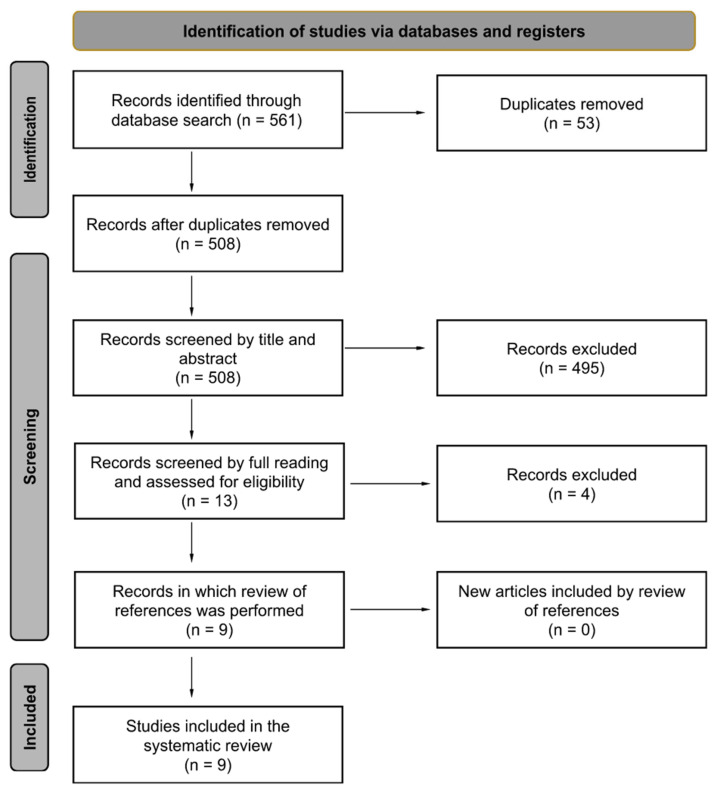
Flow diagram depicting the screening process performed.

**Figure 2 ijms-25-04912-f002:**
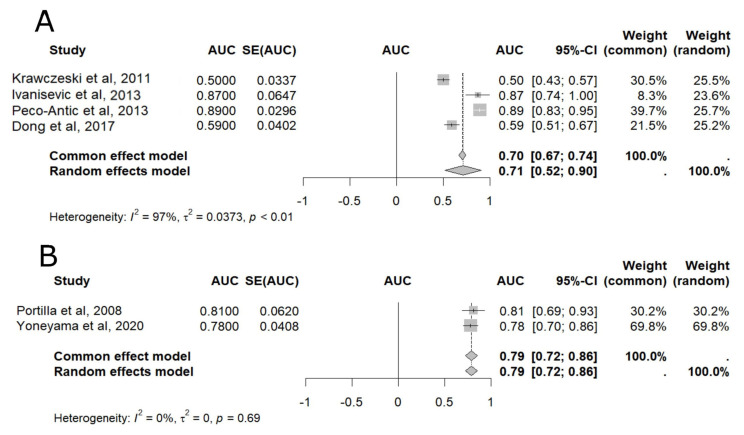
Meta-analysis of the area under the ROC curve (AUC) values of studies that evaluated the accuracy of urinary liver-type fatty acid-binding protein (L-FABP) in detecting acute kidney injury (AKI) in children after cardiopulmonary bypass (CPB). (**A**) Two-hour meta-analysis; (**B**) Four-hour meta-analysis. 95% CI: 95% confidence interval; AUC: area under the ROC curve; SE: standard error of the mean (see [16,18,19,21,22,24]).

**Figure 3 ijms-25-04912-f003:**
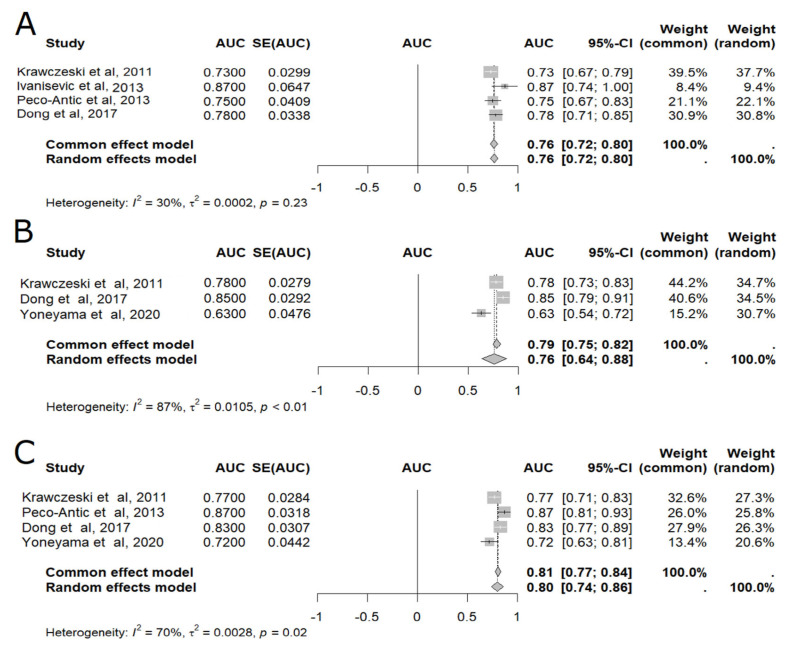
Meta-analysis of the area under the ROC curve (AUC) values of studies that evaluated the accuracy of plasmatic liver-type fatty acid-binding protein (L-FABP) in detecting acute kidney injury (AKI) in children after cardiopulmonary bypass (CPB). (**A**) Six-hour meta-analysis; (**B**) Twelve-hour meta-analysis; (**C**) Twenty four-hour meta-analysis. 95% CI: 95% confidence interval; AUC: area under the ROC curve; SE: standard error of the mean (see [16,18,21,22,24]).

**Table 1 ijms-25-04912-t001:** Characteristics of location, interventions, and sample sizes of the included studies.

Author and Year	Country	Study Design	Number of Participants	AKI Definition	Population Setting (Comparison)	Sample Sizes		AKI Severity Definition	Primary Endpoint	uL-FABP Measurements Post-CPB (Hours)
						AKI (%)	Non-AKI (%)			
Portilla et al., 2008 [19]	United States	Prospective Cohort	40	KDIGO	AKI vs. Non-AKI	21 (53%)	19 (47%)	Not specified	AKI development	0, 4, 12
Krawczeski et al., 2011 [21]	United States	Prospective Cohort	220	KDIGO	AKI vs. Non-AKI	60 (27%)	160 (73%)	pRIFLE	AKI development	0, 2, 6, 12, 24
Ivanišević et al., 2013 [18]	Serbia	Case-control	27	KDIGO	AKI vs. Non-AKI	11 (41%)	16 (59%)	pRIFLE	AKI development	0, 2, 6, 24, 48
Parikh et al., 2013 [20]	United States	Prospective Cohort	311	*	AKI vs. Non-AKI	53 (17%)	258 (83%)	pRIFLE	AKI development	0–6, 6–12, daily up to five days ***
Peco-Antić et al., 2013 [16]	Serbia	Prospective Cohort	112	**	AKI vs. Non-AKI	18 (16%)	94 (84%)	pRIFLE	AKI development	0, 2, 6, 24, 48
Zappitelli et al., 2015 [23]	Canada	Prospective Cohort	287	KDIGO	AKI vs. Non-AKI	125 (43%)	162 (57%)	KDIGO	AKI development	0–6, daily up to five days ***
Dong et al., 2017 [22]	United States, China	Prospective Cohort	150	KDIGO	AKI vs. Non-AKI	50 (33%)	100 (67%)	KDIGO	AKI development	0, 2, 6, 12, 24
Greenberg et al., 2018 [17]	United States	Prospective Cohort	408	KDIGO	AKI with progression vs. AKI without progression	176 (43%)	232 (57%)	KDIGO	AKI progression	****
Yoneyama et al., 2020 [24]	Japan	Prospective Cohort	103	KDIGO	AKI vs. Non-AKI	47 (46%)	56 (54%)	KDIGO	AKI development	0, 4, 12, 24 *****

* AKI is defined as doubling of SCr or the need for acute dialysis. ** AKI is defined as a 25% decrease of the glomerular filtration rate estimated by Schwartz’s formula. *** The article does not furnish specific time points, only time intervals: 0–6, 6–12, day 2, day 3, day 4, day 5. **** For this study, biomarkers were measured on the initial day that the serum creatinine first attained the stage 1 AKI. ***** The article utilizes time after ICU admission as the reference point for biomarker dosing. AKI, acute kidney injury; Non-AKI, no acute kidney injury; KDIGO, Kidney Disease Improving Global Outcomes; SCr, serum creatinine; pRIFLE, pediatric-modified Risk, Injury, Failure, Loss, and End-Stage Kidney Disease; uL-FABP, urinary liver-type fatty acid binding protein; CPB, cardiopulmonary bypass.

**Table 2 ijms-25-04912-t002:** Baseline study population characteristics.

Author and Year	Number of Participants	Age (Months)		Gender (Male)		Bypass Time (Minutes)		RACHS ≤ 3		Death or Dialysis	
		AKI	Non-AKI	AKI (%)	Non-AKI (%)	AKI	Non-AKI	AKI (%)	Non-AKI (%)	AKI (%)	Non-AKI (%)
Portilla et al., 2008 [19]	40	32.4 ± 9.6	51.6 ± 15.6	9 (42.9)	12 (63.2)	145 ± 12	82 ± 10	**	**	**	**
Krawczeski et al., 2011 [21]	220	39.6 (6.0–72.0)	7.2 (4.8–21.6)	26 (43)	84 (53)	113 (84–172)	92 (67–127)	57 (95)	148 (93)	1 (2)	0 (0)
Ivanišević et al., 2013 [18]	27	3.8 (0.4–28.6)	11.9 (6.1–47.4)	7 (63.6)	8 (50)	240 ± 57	119 ± 78	10 (91)	16 (100)	4 (36)	1 (6)
Parikh et al., 2013 [20]	311	8.4 (4.8–44.4)	34.8 (6.0–67.2)	27 (51)	144 (55.8)	130 (95–203)	88 (64–122)	46 (87)	251 (98)	***	***
Peco-Antić et al., 2013 [16]	112	8.4 (0.4–37.2)	21.6 (7.2–56.4)	12 (66.7)	53 (56.4)	237 (155–288)	98 (51–170)	15 (83)	90 (96)	7 (39)	0 (0)
Zappitelli et al., 2015 [23]	287	30.0 ± 42.2	57.6 ± 56.4	65 (52)	60 (48)	130 (72)	90 (49)	114 (91)	159 (98)	4 (3)	0 (0)
Dong et al., 2017 [22]	150	60.2 ± 10.9	17.4 ± 15.5	20 (40)	57 (57)	133 (116–153)	88 (80–97)	44 (88)	96 (96)	0 (0)	0 (0)
Greenberg et al., 2018 [17] *	408	22.8 ± 34.8	32.4 ± 48.0	16 (57.1)	81 (54.7)	142 ± 55	116 ± 65	26 (93)	138 (94)	4 (15)	4 (3)
Yoneyama et al., 2020 [24]	103	36.0 ± 61.2	69.6 ± 80.4	25 (53.1)	29 (51.7)	231.4 ± 129.4	126.1 ± 65.0	39 (83)	51 (91)	**	**

* The article categorizes the results into two groups: AKI with progression and AKI without progression. ** The paper does not furnish this information. *** The authors present this outcome by categorizing patients into quintiles based on urinary L-FABP and KIM-1 levels, rather than by AKI and Non-AKI classifications. AKI, acute kidney injury; Non-AKI, no acute kidney injury; RACHS, Risk adjustment in congenital heart surgery; L-FABP, Liver Fatty Acid-Binding Protein; KIM-1, Kidney Injury Molecule-1.

**Table 3 ijms-25-04912-t003:** AUC of uL-FABP to diagnose AKI at different post-CPB timepoints.

Author and Year	0 h	2 h	4 h	6 h	12 h	24 h
Portilla et al., 2008 [19]	*	*	0.81	*	*	*
Krawczeski et al., 2011 [21]	*	0.50	*	0.73	0.78	0.77
Ivanišević et al., 2013 [18]	*	0.87	*	0.87	*	*
Peco-Antić et al., 2013 [16]	*	0.89	*	0.75	*	0.87
Dong et al., 2017 [22]	*	0.59	*	0.78	0.85	0.83
Yoneyama et al., 2020 [24]	0.82	*	0.78	*	0.63	0.72

Parikh et al., 2013 [20] and Zappitelli et al., 2015 [23] were excluded from the table as they only provided the AUC within time intervals of 0–6 h and 6–12 h without specific timepoints. However, information regarding this paper is available in the text. Greenberg et al., 2018 [17] was excluded due to its focus on predicting the progression of AKI rather than its diagnosis, and it also lacked specific information regarding the timing of AUC evaluation. * The AUC for this timepoint is not available in the paper.

## Data Availability

Not applicable.

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
