# Peer review of "Urinary L-FABP as an Early Biomarker for Pediatric Acute Kidney Injury Following Cardiac Surgery with Cardiopulmonary Bypass: A Systematic Review and Meta-Analysis"

_ijms, 2024, doi:10.3390/ijms25094912_

Round 1
Reviewer 1 Report
Comments and Suggestions for Authors
This a well written and comprehensive review of post CP bypass acute renal injury. The methodology and discussion are appropriate.
My only recommendation would be to place LFABP in context of renal pathophysiology and provide more background into why this was chosen.
Comments on the Quality of English Language
no issues
Author Response
Answer to Reviewer #1 comments and action taken:
Reviewer #1:
This a well written and comprehensive review of post CP bypass acute renal injury. The methodology and discussion are appropriate.
My only recommendation would be to place LFABP in context of renal pathophysiology and provide more background into why this was chosen.
Author’s reply: We would like to thank Reviewer #1 for the comment and suggestion. In order to clarify L-FABPs’ role both in the normal kidney and during acute injury, as well as to better describe the rationale for its investigation as an early biomarker in AKI, we added the following (page 2, lines 73-83):
“In the kidneys, the L-FABP is mostly expressed in the proximal tubules, where it is responsible for the transportation of fatty acids into the tubular epithelial cells’ mitochondria to be converted into energy[9,10]. Especially in the kidneys, the L-FABP gene has been found to be upregulated during hypoxic stress and ischemia-reperfusion injury, two major mechanisms involved in the pathogenesis of AKI-CPB[9,10]. During a hypoxic event, the damage of the proximal tubules is largely caused by the accumulation of reactive oxygen species (ROS) in the kidneys. This process is inhibited by L-FABPs’ capacity to bind with lipid peroxidation products and induce the excretion of these molecules in the urine, being detected as urinary L-FABP (uL-FABP)[10]. Therefore, evidence indicates that uL-FABP undergoes a compensatory increase very early in hypoxia-mediated AKI.”

Reviewer 2 Report
Comments and Suggestions for Authors
Does the introduction provide sufficient background and include all relevant references?
· Should be adjusted, based on issued elaborated below, especially in the section of design.
· Introduction should contain the information about the uL-FABP analysis methods and their impact (or rather absence) on results, if they are variable. Alternatively, this might be moved to discussion section as well.
Are all the cited references relevant to the research?
· Should be improved. All references in the text are written without brackets (e.g., line 360). This makes text reading difficult
· Some references are not informative as per any agreed system – i.e., #13 and #14 should provide page numbers
Is the research design appropriate?
· Must be clarified and amended. The authors state the purpose of this study that is broader than the design selected. The purpose of this study was to evaluate the potential value of uL-FABP in early predicting AKI among infants submitted to cardiac surgery with cardiopulmonary bypass. The design however was suitable for diagnostic BM and not predictive biomarkers (see https://www.ncbi.nlm.nih.gov/books/NBK338448/ )
· The search strategy is briefly described in section 2.3, although search equations via different libraries are not provided
· The search strategy did not contain incident AKI criteria according to RIFLE. This is one of significant omissions of the study.
· The quality of evidence was assessed using not the GRADE system, but by the JBI Critical appraisal checklist for diagnostic test accuracy studies. This is not sufficiently informative quality assessment system and should be complemented by the GRADE system
· The accuracy should be described not only as is proposed by the authors (area under the ROC curve (AUC) for the accuracy of AKI detection by uL-FABP) but also as (TP + TN) / (TP + FN + FP + TN).
· Ther results should be calculated for sensitivity, specificity, likelihood rations with positive (LH(+) and negative (LR(-)) and diagnostic odds rations (DOR) together with 95% CI
Are the methods adequately described?
· Must be clarified and amended. See also comments for study design.
· The detailed PROSPERO protocol registration information is missing. The review report should be provided
· The authors describe the missing information for statistical analysis (line 137) but methods where and how this was mitigated is provided. This information should be provided and rationale for the method justified.
· In the screening process diagram the qualitative and quantitative syntheses should be separated.
· Qualitative description of the studies is not detailed enough
o Author and year should be complemented with country and reference
o Population setting should be mentioned in a separate column
o Study design should be mentioned in a separate column
o Primary endpoint and secondary endpoints should be distinguished
o Cut-off values used in the studies should be clearly mentioned (this might be provided in supplement
o AKI criteria should be specified as per AKIN, KDIGO, RIFLE or mixture and what parameters encountered – creatinine only or urine output only or both
o Total number of patients should be included in separate column
o AKI severity should be mentioned in a separate column
o Timings of measurement should be mentioned in a separate column
· Formulas used to complete the datasets in case of missing data should be provided.
Are the results clearly presented?
· Must be clarified and amended as per study design and methods.
· Comparability of baseline results between the studies should be described
· Comparability of peak and uL-FABP results between the studies should be described in more details (line 290). Similarly, and levels that could “predict” should be specifically mentioned (i.e., line 312, 317, 322)
· It seems that intention was to mention urinary L-FABP, not plasmatic (in line 348)
Are the conclusions supported by the results?
· To early to address at this stage and must be clarified. Conclusions could be assessed after previous methodological issues are addressed.
Author Response
Answer to Reviewer #2 comments and action taken:
Reviewer #2:
Does the introduction provide sufficient background and include all relevant references?
· Should be adjusted, based on issued elaborated below, especially in the section of design.
· Introduction should contain the information about the uL-FABP analysis methods and their impact (or rather absence) on results, if they are variable. Alternatively, this might be moved to discussion section as well.
Author’s reply: We would like to thank Reviewer #2 for the comments and suggestions. In all the studies, uL-FABP measurements were consistently conducted using the Enzyme-Linked Immunosorbent Assay (ELISA) technique, a widely utilized method for assessing uL-FABP levels in the literature. This information was included in the introduction and also described in the results section.
Introduction (page 2, lines 87-88): "Therefore, the assessment of L-FABP levels, commonly conducted through the Enzyme-Linked Immunosorbent Assay (ELISA) technique, emerges as a potential strategy for the prediction and early diagnosis of AKI post-CPB."
Results (page 7, lines 198-200): "In all the studies, L-FABP measurements were consistently conducted using the Enzyme-Linked Immunosorbent Assay (ELISA) technique."
Are all the cited references relevant to the research?
· Should be improved. All references in the text are written without brackets (e.g., line 360). This makes text reading difficult. Some references are not informative as per any agreed system – i.e., #13 and #14 should provide page numbers
Author’s reply: We would like to thank Reviewer #2 for the comments and suggestions. In the original submitted manuscript, references were cited in the text as superscript number and according to Journals’ format requirements. Unfortunately, it seems that the references lost correct format, making the manuscript difficult to read and navigate. As instructed, all references were rewritten using brackets, and were adjusted to the appropriate formatting. These alterations are highlighted in yellow.
Is the research design appropriate?
· Must be clarified and amended. The authors state the purpose of this study that is broader than the design selected. The purpose of this study was to evaluate the potential value of uL-FABP in early predicting AKI among infants submitted to cardiac surgery with cardiopulmonary bypass. The design however was suitable for diagnostic BM and not predictive biomarkers (see https://www.ncbi.nlm.nih.gov/books/NBK338448/ )
Author’s reply: We would like to thank Reviewer #2 for the comments and suggestions. In our study, uL-FABP was evaluated both as an early diagnostic biomarker and as a prognostic molecule, aiming to predict the progression of CPB-AKI to more severe stages. Therefore, we partially agree with the commentary above. Aiming to clarify this issue, we changed the term “predict” to “early diagnose” (or appropriate synonyms) in all text sections that referred to the uL-FABP’s capacity to identify CPB-AKI. These changes are highlighted in yellow.
· The search strategy is briefly described in section 2.3, although search equations via different libraries are not provided
Author’s reply: We would like to thank Reviewer #2 for the comments and suggestions, which, we believe, were made to improve our manuscript. We designed a table with detailed information about the search strategy and it is now available in the Supplementary Table 1. Also, the following segment was added to the main manuscript (page 3, lines 120-121): ”The complete search strategy can be found in Supplementary Table 1.”
· The search strategy did not contain incident AKI criteria according to RIFLE. This is one of significant omissions of the study.
Author’s reply: We would like to thank Reviewer #2 for the comments and suggestions. We designed our search strategy to be composed of Medical Subject Headings (MeSH) terms and, since “Acute Kidney Injury” is a MeSH term and it is included in our search strategy, any search that include this keyword will also contemplate all its MeSH-identified synonyms, including studies comprising patients with AKI defined by the RIFLE criteria. Additionally, since RIFLE is not a MeSH term, it was not included in the search strategy.
· The quality of evidence was assessed using not the GRADE system, but by the JBI Critical appraisal checklist for diagnostic test accuracy studies. This is not sufficiently informative quality assessment system and should be complemented by the GRADE system.
Author’s reply: We would like to thank Reviewer #2 for the comments and suggestions. However, the JBI Critical Appraisal Checklist is recognized for its reliability and informative quality assessment. Notably, it is endorsed as the quality assessment tool with the widest range of applicability by recent studies regarding the topic (https://doi.org/10.1186/s40779-020-00238-8; https://doi.org/10.11124/JBIES-23-00139). We believe that the use of JBI Critical Appraisal Checklist is sufficient and should not be complemented by the GRADE system.
· The accuracy should be described not only as is proposed by the authors (area under the ROC curve (AUC) for the accuracy of AKI detection by uL-FABP) but also as (TP + TN) / (TP + FN + FP + TN).
Author’s reply: We would like to thank Reviewer #2 for the comment and suggestion. Unfortunately, the studies evaluated in our systematic review and meta-analysis do not disclose or establish cut-off values employed for the early diagnosis of CPB-AKI at different time-points, with the AUC being the most extensively described metric in all included papers. The absence of specific cut-off values hinders the calculation of accuracy-related metrics, such as (TP + TN) / (TP + FN + FP + TN), other than AUC. We, nevertheless, agree that the absence of this information represents a limitation of our study. To include this limitation, we have added the following in the manuscript (page 17, lines 472-477):
“Additionally, despite the AUC consisting in an important and reliable metric for the evaluation of a biomarker’s diagnostic and predictive capacity, an in-depth assessment of a diagnostic test also relies on the analysis of statistical properties associated with a fixed cut-off value, such as accuracy and likelihood ratios. Since the included studies did not establish cut-off values for the early diagnosis of post-CPB AKI, these metrics were not evaluated.”
· The results should be calculated for sensitivity, specificity, likelihood rations with positive (LH(+) and negative (LR(-)) and diagnostic odds rations (DOR) together with 95% CI
Author’s reply: We would like to thank Reviewer #2 for the comment and suggestion. As stated in the previous comment, the included articles did not establish or implement cut-off values for the early diagnose of AKI-CPB, a fact that hinders the evaluation of statistical properties associated with a fixed cut-off value, such as sensitivity, especificity, likelihood ratios and diagnostic odds ratios. To include this limitation, we have added the following in the manuscript (page 17, lines 472-477):
“Additionally, despite the AUC consisting in an important and reliable metric for the evaluation of a biomarker’s diagnostic and predictive capacity, an in-depth assessment of a diagnostic test also relies on the analysis of statistical properties associated with a fixed cut-off value, such as accuracy and likelihood ratios. Since the included studies did not establish cut-off values for the early diagnosis of post-CPB AKI, these metrics were not evaluated.”
Are the methods adequately described?
· Must be clarified and amended. See also comments for study design.
· The detailed PROSPERO protocol registration information is missing. The review report should be provided
Author’s reply: We would like to thank Reviewer #2 for the comments and suggestions, which, we believe, were made to improve our manuscript. The detailed PROSPERO protocol registration was made available by inserting the link to the full document, and the following segment was added to the text (page 3, line 96-100): “under the title ‘Biomarkers’ effectiveness to define diagnosis, prognosis and severity of pediatric acute kidney injury in Emergency Units: a systematic review”. The registration can be found in (https://www.crd.york.ac.uk/prospero/display_record.php?ID=CRD42022318748).”
· The authors describe the missing information for statistical analysis (line 137) in the methods where and how this was mitigated is provided. This information should be provided and rationale for the method justified.
Author’s reply: We would like to thank Reviewer #2 for the comments and suggestions, which, we believe, improved our manuscript. For performing the analysis, we only used information that was presented in the included studies. In case of missing enough information for analysis, the analysis was not performed, to avoid misinterpretation and inadequate speculation. We only calculated additional statistical parameters when there was enough information for calculating it. For instance, the standard statistical error was calculated by using the standard deviation and the total number of patients, as described by Van Belle et al. (2004) and in other numerous books in biostatistics. Therefore, we did not describe missing information — as we only used information which was presented in the included studies. To make this clearer, we added the following information in the manuscript (page 7, lines 168 and 169):
“The SE and the 95%CI were estimated as described by Van Belle et al. (2004).”
We have, therefore, also included the following reference:
Fisher, L. D., & Van Belle, G. (1997). Biostatistics: A methodology for the health sciences. Biometrics, 53(3), 1182. https://doi.org/10.2307/2533583
· In the screening process diagram the qualitative and quantitative syntheses should be separated.
Author’s reply: We would like to thank Reviewer #2 for the comments and suggestions, which, we believe, were made to improve our manuscript. Our flow diagram was crafted following the PRISMA 2020 model, chosen for its adherence to the guidelines set forth in the PRISMA statement. We intentionally adopted this approach to ensure that the reasons for inclusion and exclusion are comprehensively depicted with the quantitative analysis within a single visual representation.
We have noticed that the PRISMA guideline is referenced as its older version (reference #12). We have, therefore, substitute reference #12 by the newer PRISMA 2020 guideline reference, which is as follows:
Page MJ, McKenzie JE, Bossuyt PM, Boutron I, Hoffmann TC, Mulrow CD, et al. The PRISMA 2020 statement: An updated guideline for reporting systematic reviews. British Medical Journal. 2021 Mar 29;372(71).
· Qualitative description of the studies is not detailed enough
o Author and year should be complemented with country and reference
o Population setting should be mentioned in a separate column
o Study design should be mentioned in a separate column
o Primary endpoint and secondary endpoints should be distinguished
o Cut-off values used in the studies should be clearly mentioned (this might be provided in supplement
o AKI criteria should be specified as per AKIN, KDIGO, RIFLE or mixture and what parameters encountered – creatinine only or urine output only or both
o Total number of patients should be included in separate column
o AKI severity should be mentioned in a separate column
o Timings of measurement should be mentioned in a separate column
Author’s reply: We would like to thank Reviewer #2 for the comments and suggestions, which, we believe, were made to improve our manuscript. We have included references to each paper in all tables. While we did mention the study design and country of production in Table 1, we found that incorporating this information into Table 2 would overcrowded the table and potentially lead to confusion due to the number of existing columns. Therefore, we believe it is more appropriate to retain the mention of the study design and country of production solely in Table 1. AKI criteria and AKI severity were described in separate columns. Study design is, as required, now mentioned in a separate column in Table 1.
The primary outcome was added as a separate column in Table 1, and also briefly described in the text. Since none of the included studies clearly defined the analyzed secondary outcomes, we did not include a separate column for this variable. We also added the following segment to the main manuscript (page 7, lines 196-201): “The primary outcome was the development of post-CPB AKI in 8 of the 9 (88.9%) included studies[15,17-23], with one study evaluating AKI progression as the primary outcome[16]. None of the included studies clearly defines the secondary outcomes analyzed. However, since clinically relevant outcomes (hospital length-of-stay, progression to dialysis, and death) other than AKI development were reported, these results were also qualitatively analyzed in the results section.”
We have also included the total number of patients in Table 2 to provide additional insight into the study size. This enhancement aims to offer a more comprehensive understanding of the data presented in Table 2. Timings of measurement are now available in a separate column in table 1. The studies evaluated in our review do not disclose or establish cut-off values employed for the early diagnosis of CPB-AKI at different time-points, with the AUC being the most extensively described metric in all included papers, so we do not see fit to include this parameter in our table.
· Formulas used to complete the datasets in case of missing data should be provided.
Author’s reply: We would like to thank Reviewer #2 for the comments and suggestions, which, we believe, were made to improve our manuscript. For performing the analysis, we only used information that was presented in the included studies. In case of missing enough information for analysis, the analysis was not performed, in order to not perform and present inadequate information. When the standard error was not provided, but there was enough information for calculating it using the standard deviation and the total number of patients, as described by Van Belle et al. (2004) and in other numerous books in biostatistics. Therefore, we did not describe missing information — as we only used information which was presented in the included studies. To make this clearer, we added the following information in the manuscript (page 7, lines 169 and 170):
“The SE and the 95%CI were estimated as described by Van Belle et al. (2004).”
We have, therefore, also included the following reference:
Van Belle, G., Fisher, L. D., Heagerty, P. J., & Lumley, T. Biostatistics: a methodology for the health sciences. John Wiley & Sons. 2004.
Are the results clearly presented?
· Must be clarified and amended as per study design and methods.
· Comparability of baseline results between the studies should be described
Author’s reply: We would like to thank Reviewer #2 for the comments and suggestions. We agree that analyzing baseline results holds clinical significance only when the samples are obtained post-surgical intervention. Baseline data collected before the surgical procedure may not accurately reflect exposure to the causal factor. Therefore, we have opted to include in our analysis only those baseline results acquired immediately after surgery or upon post-CPB ICU admission. This refinement ensures a more precise evaluation of the impact of the surgical procedure itself. This information was included in the results section (page 7, lines 197-198):
"Baseline results were only evaluated when they were defined as the time immediately after surgery or at post-CPB ICU admission."
Additionally, we excluded results pertaining to baseline measurements obtained before the surgical procedure, as well as the meta-analysis of this time-point.
· Comparability of peak and uL-FABP results between the studies should be described in more details (line 290). Similarly, and levels that could “predict” should be specifically mentioned (i.e., line 312, 317, 322).
Author’s reply:
We would like to thank Reviewer #2 for the comments and suggestions. As reported in a previous comment and aiming to improve clarity, we changed the term “predict” to “early diagnose” (or appropriate synonyms) in all text sections that referred to the uL-FABP’s capacity to identify CPB-AKI; these changes are highlighted in yellow. Additionally, we added an additional explanation to the definition of “peak uL-FABP”, in order to more clearly define the adopted concept. The added segment is as follows (page 12, lines 326-327): ”i.e. its highest value in 24 hours.”
· It seems that intention was to mention urinary L-FABP, not plasmatic (in line 348).
Author’s reply: We would like to thank Reviewer #2 for the comment. The word “plasmatic” was replaced by “urinary”. The change can be seen on page 14, line 385.
Are the conclusions supported by the results?
· To early to address at this stage and must be clarified. Conclusions could be assessed after previous methodological issues are addressed.
Author’s reply: We would like to thank Reviewer #2 for the comments and suggestions, which, we believe, were made to improve our manuscript.

Round 2
Reviewer 2 Report
Comments and Suggestions for Authors
Authors made substantial changes and amendments and now the text is fairly representing scientific level it delivers.
The value of main message about potential for earlier AKI detection is scientifically sound although without any practical value, as no cut-off values are described and discussed. I would see very important to improve the article by including one short paragraph about that (in discussion section or introduction)